# Intracellular morphogenesis of diatom silica is guided by local variations in membrane curvature

Lior Aram [1,8], Diede de Haan [1,8], Neta Varsano[2], James B. Gilchrist [3], Christoph Heintze[4], Ron Rotkopf [5], Katya Rechav [2], Nadav Elad [2], Nils Kröger [4,6,7] & Assaf Gal [1] ✉

Silica cell-wall formation in diatoms is a showcase for the ability of organisms to control inorganic mineralization. The process of silicification by these unicellular algae is tightly regulated within a membrane-bound organelle, the silica deposition vesicle (SDV). Two opposing scenarios were proposed to explain the tight regulation of this intracellular process: a template-mediated process that relies on preformed scaffolds, or a template-independent self-assembly process. The present work points to a third scenario, where the SDV membrane is a dynamic mold that shapes the forming silica. We use in-cell cryo-electron tomography to visualize the silicification process in situ, in its native-state, and with a nanometer-scale resolution. This reveals that the plasma membrane interacts with the SDV membrane via physical tethering at membrane contact sites, where the curvature of the tethered side of the SDV membrane mirrors the intricate silica topography. We propose that silica growth and morphogenesis result from the biophysical properties of the SDV and plasma membranes.

The hallmark of the silica cell walls of diatoms is their species-specific geometrical patterning. The intricate architectural features span the length scale from nanometers up to hundreds of micrometers[1,2]. The fact that these unicellular organisms can biosynthesize such complex architectures makes them prime examples for the biologically controlled morphogenesis of minerals[3]. In the past, microscopy studies revealed a detailed picture of intermediate silica morphogenesis stages and cellular events during silica biogenesis[4,5]. More recently, biochemical and genetic tools have started to identify an increasing cohort of biomolecules involved in diatom cell wall formation[6–8]. Nevertheless, how all these biomolecules are integrated into a functional infrastructure to orchestrate silica morphogenesis has remained largely elusive.

A diatom cell wall consists of two types of building blocks: two plate-shaped valves and several ring-shaped girdle bands. The formation of an individual silica element takes place inside a dedicated intracellular compartment called a silica deposition vesicle (SDV)[9,10]. Various hypotheses regarding the mechanism of SDV-mediated silica morphogenesis were proposed and can be broadly divided into template-mediated and template-independent processes. The *template-mediated mechanisms* rely on self-assembled organic scaffolds that exhibit nanoscale patterns over microscale dimensions. The assembly of the scaffold is assumed to occur inside the SDV lumen (internal patterning) or on the cytoplasmic surface of the SDV membrane (external patterning). In external patterning, cytoskeleton fibers

[1]Department of Plant and Environmental Sciences, Weizmann Institute of Science, Rehovot, Israel. [2]Department of Chemical Research Support, Weizmann Institute of Science, Rehovot, Israel. [3]Electron Bio-Imaging Centre, Diamond Light Source, Harwell Science and Innovation Campus, Didcot, United Kingdom. [4]B CUBE – Center for Molecular Bioengineering, Technische Universität Dresden, Dresden, Germany. [5]Life Sciences Core Facilities, Weizmann Institute of Science, Rehovot, Israel. [6]Cluster of Excellence Physics of Life, Technische Universität Dresden, Dresden, Germany. [7]Faculty of Chemistry and Food Chemistry, Technische Universität Dresden, Dresden, Germany. [8]These authors contributed equally: Lior Aram, Diede de Haan. ✉e-mail: assaf.gal@weizmann.ac.il

(microtubules, actin), mitochondria, and so-called 'spacer vesicles' are believed to mold the SDV membrane into species-specific shapes, including regularly arranged indentations that guide the formation of the biosilica patterns[1,2,11,12]. In internal patterning, biomolecules in the SDV lumen or on its membrane are suggested to form a microscale organic matrix that defines nanoscale patterns for silica formation[6,13–16].

The *template-independent mechanisms* hypothesize that silica morphogenesis is controlled by fluxes of soluble silica precursors and interacting macromolecules into the SDV. Within the SDV, the self-assembly of these components is controlled by their diffusion rate and the rates of the chemical reaction cascades that convert the soluble silica precursors into solid silica. Mathematical modeling of such processes based on diffusion-limited aggregation or reaction-diffusion systems successfully replicates certain sub-features of the biosilica morphology[14,17–20]. To date, there is no consensus on how diatoms control silica morphogenesis since both the template-mediated and the template-independent mechanisms are far from providing a comprehensive explanation. This is a consequence of our incomplete knowledge of the biochemical composition of the SDV and the physico-chemical conditions inside it. Furthermore, there is no information on the native structure of the SDV and its relationship to the nascent silica structures inside it and cytoplasmic structures around it.

The developments in cryo-electron microscopy now allow the collection of high-resolution structural data within cells that are vitrified without any chemical fixation or staining and thus are in a native-like state[21,22]. This suite of techniques already provided detailed information on mineral formation by organisms[23,24], making it a promising avenue to study silica formation within the SDV. The diatom *Thalassiosira pseudonana* has emerged as the model species for studying the molecular basis of silica morphogenesis[25], and thus we used cryo-electron tomography (cryoET) to obtain near native images of *T. pseudonana* valve SDVs throughout all stages of silica morphogenesis. We note that *T. pseudonana* was re-named *Cyclotella nana*[25], but in order to be consistent with previous literature, we will use the previous name. The aim of the current study was to unveil in unprecedented detail the internal structure of the valve SDV and the relationship of the SDV membrane with both nascent silica and other subcellular components.

## Results

We synchronized the cell cycle in *T. pseudonana* cultures by a silicon (Si) starvation period that causes the cells to arrest their growth just before cell division[26]. Replenishing silicon to the growth medium induced a synchronized progress of the cell cycle, and after three hours, the cell population was enriched with cells at the stage of valve formation[26]. To prepare samples for cryoET, synchronized cells were vitrified without any chemical fixation or staining and then thinned to ~200 nm lamellae using a cryo-focused-ion-beam instrument (Fig. 1A–C). The barrel-shaped geometry of *T. pseudonana* plays an important role in the process of lamella preparation, as the cells lay preferentially on their girdle bands, with the valves (mature and forming), perpendicular to the substrate. This implies that lamellae prepared from the middle of the cell will include a central slice of the SDV (Fig. 1C). Therefore, data collected from many lamellae contain all SDV parts, from its central section to the growing front, and the collected sub-volumes of many SDVs are demonstrative for the entire valve SDV.

We collected cryoET data from 47 cell pairs shortly after cell division (see "Methods" section). Each lamella usually contained two daughter cells, still enveloped by the parental cell wall (Fig. 1D). Thirty-six high-resolution datasets were collected from the region where the two daughter cells are adjacent. In this region, each cell develops a valve SDV proximal to the plasma membrane. Identifying an SDV proved straightforward as its flat morphology and electron-dense silica content make it distinct from any other cellular structure (Fig. 1E and Supplementary Fig. S1). Since synchronization is not perfect, the vitrified cells included various stages of valve formation, from nascent valve SDVs to already exocytosed valves.

The developmental sequence of the valve was previously established by extracting forming valves from their SDVs[13,27]. Valve formation starts with a central annulus, from which radial silica ribs emanate and branch towards the valve circumference (Fig. 1F). At a later stage, the gaps between the ribs are filled with a layer perforated with nanoscale openings (cribrum pores; Fig. 1B). We created a derived timeline of valve formation within the SDV by arranging the collected datasets in increasing order of SDV diameter, silica thickness, and development of secondary ornaments (Fig. 2 and Supplementary Figs. S2, S3). Even though this timeline cannot be unequivocally defined, it is robust enough to serve as a basis to separate the growth

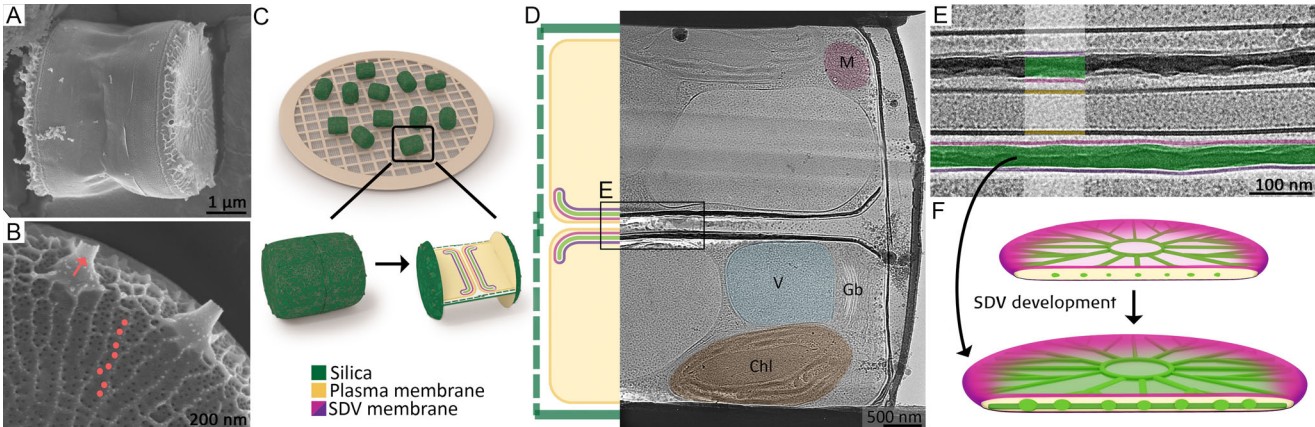

**Fig. 1 | Imaging SDVs inside *T. pseudonana* cells using cryoET.**
**A**, **B** Representative scanning electron microscopy images representing at least two samples of: (**A**) a cell lying on its girdle bands, and (**B**) higher magnification of a bleach-cleaned valve showing the cribrum pores (some indicated with red circles) and fultoportulae (red arrow). **C** A scheme of the preferred geometry of lamella preparation on vitrified cells. **D** A representative cryo-TEM image of one entire lamella, out of at least 20 lamellae, capturing two daughter cells at the stage of valve formation. The scheme on the left and artificial colors on the image are provided as a guide to the eye. M – mitochondrion, V – vacuole, Chl – chloroplast, Gb – Golgi body. **E** High magnification image from a different cell of an area analogous to the one marked in (**D**) shows the two SDVs. Some features are highlighted using the same color code. **F** Schemes of valve SDVs at two different developmental stages, showing how a cross-section in an immature SDV will contain slices of the initial silica ribs, and a section in a more developed SDV will show a continuous slab with ornaments and nano-pores.

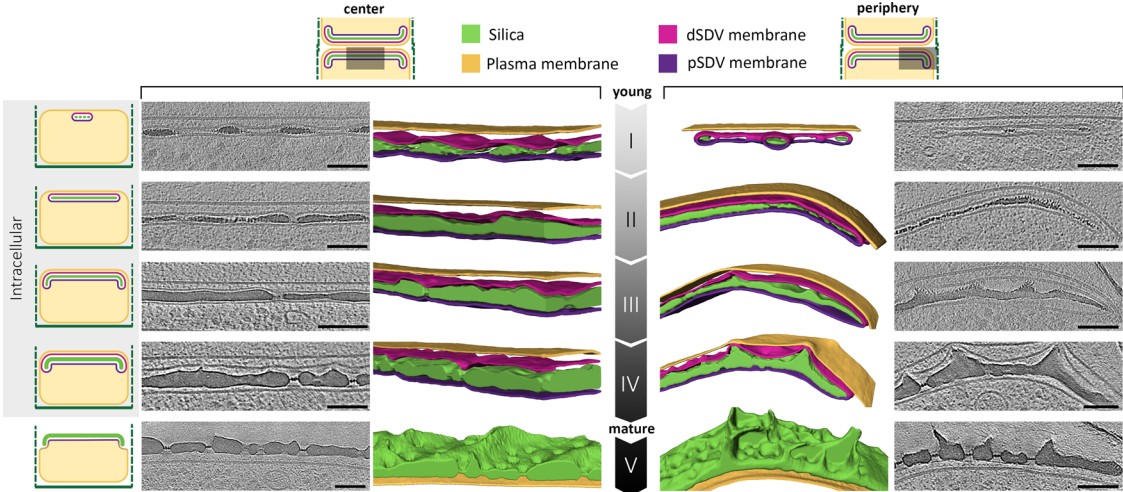

**Fig. 2 | Structural characteristics of the SDV developmental sequence imaged by in-cell cryoET.** Ten representative datasets, five taken at the SDV center (left) and five at its periphery (right), are arranged according to the degree of silicification. This shows a sequence that starts with the initial stage of distinct silica ribs (Stage I), and proceeds, via a structurally mature valve inside the SDV (Stage IV), to a recently exocytosed valve (Stage V). For each dataset, a slice through the reconstructed volume is shown next to a surface representation of the segmented cellular structures. dSDV – distal SDV, pSDV – proximal SDV. Scale bars are 100 nm. See Supplementary Figs. S2, S3 for a summary of all collected datasets.

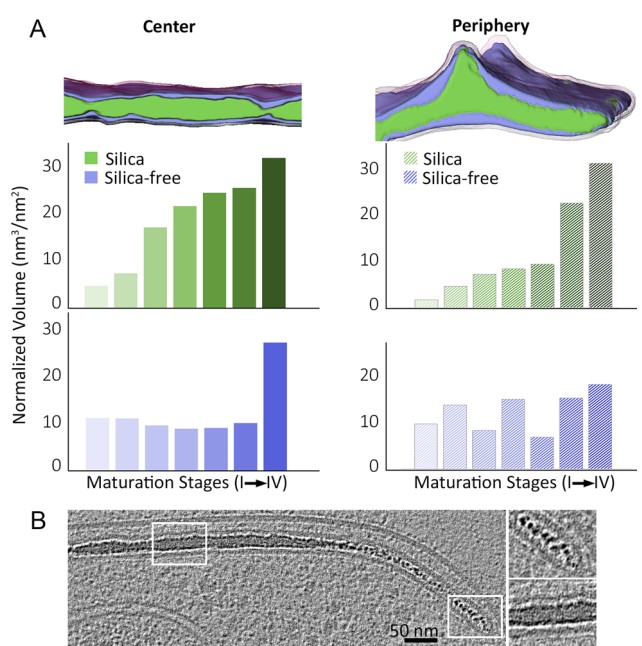

**Fig. 3 | Structural analyses of growing silica inside the SDV. A** The changes in normalized volumes of the two phases inside the SDV, the dense silica (green) and the luminal silica-free solution surrounding it (purple). Seven datasets for the SDV center and seven for its periphery were analyzed and ordered according to their developmental stage (A detailed legend is present in Supplementary Figs. S2, S3). Silica volume increases with the maturation stage (linear regression $p = 0.000187$*** for the center and $p = 0.00388$** for the periphery). The slope of the silica-free volume linear regression is not significantly different from zero (linear regression $p = 0.159021$ for the center and $p = 0.2528$ for the periphery). The last maturation stage of the silica-free volume in the SDV center was omitted from the calculation (see also Supplementary Table S1 and Supplementary Fig. S6). Source data are provided as a Source Data file. **B** A cryoET slice from Stage II in Fig. 2, with insets showing higher magnifications of two regions with different silica textures (see also Supplementary Fig. S5), this dataset is representative of 34 collected tomograms containing an SDV.

process into four representative stages, which were reconstructed for two characteristic locations inside the cell, the central region of the SDV and its growing edge (Fig. 2). The observed sequence of events is in accordance with the previous studies on *T. pseudonana* and other diatoms, demonstrating that the SDV is a flat organelle that expands laterally and that the initial silica structures are radial ribs, only 10–20 nm in thickness, which later get thicker to form the characteristic perforated silica pattern[10,13].

We analyzed the structural components within the SDV in order to investigate the feasibility of the template-independent phase separation model, where the initial SDV solution separates into a dense silica phase and a dilute residual phase[15]. For this, we segmented the voxels of silica and SDV membranes in the datasets and calculated the volumes of the dense silica phase and the non-silicified luminal solution. In order to compare SDV slices from different lamellae, the values from each dataset were normalized to an area unit of the SDV (Fig. 3A and Supplementary Fig. S4). This analysis showed that the normalized silica volume increased by a factor of ten during the entire process of valve biogenesis (Fig. 3A). In the center of the SDV, the largest differences in silica volume occurred during the initial stages of silica thickening. In the SDV periphery, silica volume was low until the stage where the SDV reached its full diameter and the silica thickened at its periphery. During the initial stages of mineralization, the silica surface had a granular texture (Fig. 3B and Supplementary Fig. S5), possibly pointing to a local and short-lived sol-gel maturation process in which silica nanoparticles sinter to a homogenous mineral phase to reduce surface energy[28,29]. In contrast to the normalized silica volume, the normalized luminal solution volume does not show a clear trend in volume change, except for the very last steps of silica formation, where the SDV membrane dilated in preparation for exocytosis (Fig. 3A, Stage IV center). These analyses demonstrate that silica growth proceeds in a highly confined space and does not involve an intermittent net increase of surrounding luminal solution as expected from a simple phase separation process. Instead, silica growth appears to be perfectly synchronized with the delivery of material for SDV expansion.

The source of the material required for SDV growth is still a matter of debate, and fusion with transport vesicles was proposed as a possible mechanism[4,30–33]. Here, we did not observe a single event of a vesicle fusing with the valve SDV, even though such events are readily visible with cryoET in other biological systems[34–36], and also in our datasets in other cellular contexts (Supplementary Fig. S7). Statistical

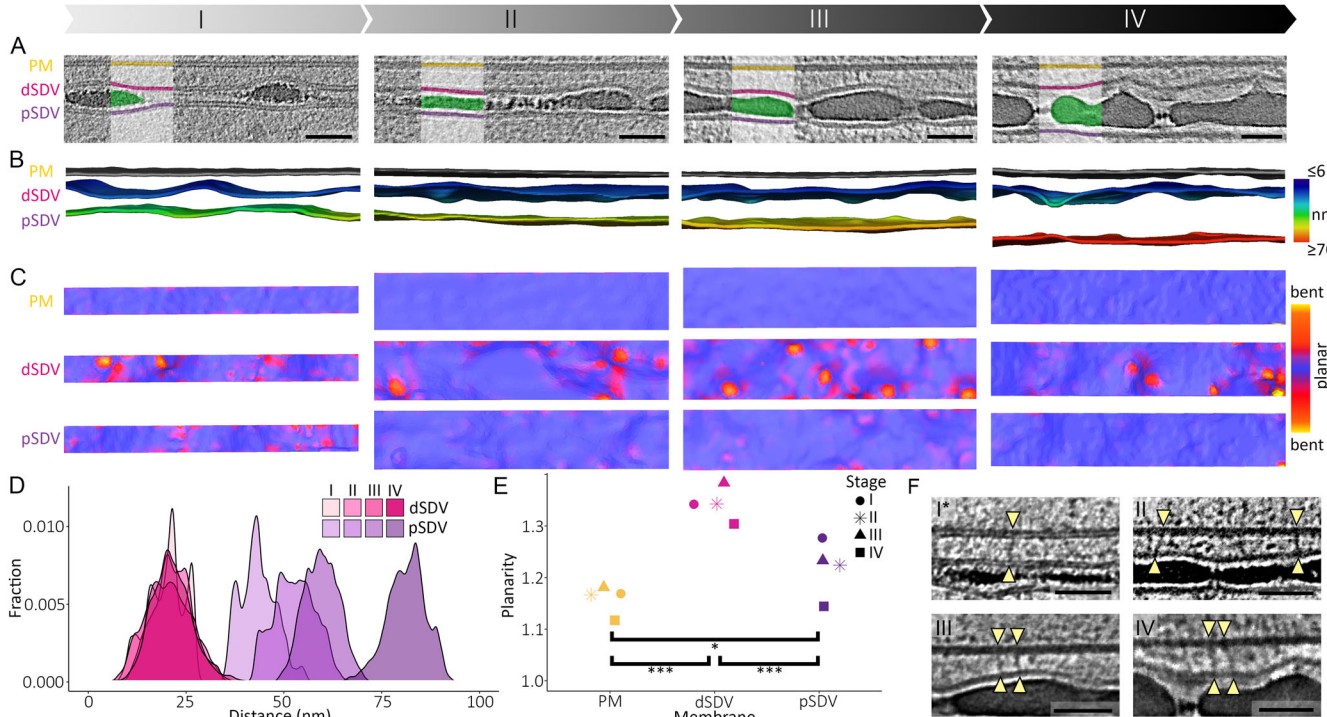

**Fig. 4 | Relationship between the plasma membrane and both sides of the SDV membrane. A** Details from the central regions of valve SDVs in stages I–IV (same as in Fig. 2) with the silica and membranes highlighted with the same color code as in Fig. 2. **B** Surface representation of membranes in the same datasets; the two sides of the SDV membranes are color-coded according to their respective distance from the plasma membrane. **C** The same three membranes for each stage are presented from a top view; the color-coding is according to the value of the local Gaussian curvature. **D** Density plots of the SDV membrane distances that are visualized in (**B**). Examining effect sizes (Cliff's Delta) shows almost complete overlap between dSDV datasets (< 0.2) and virtually no overlap between pSDV datasets (values close to 1). See also Supplementary Table S2. **E** The planarity value for each of the membranes. The total surface area of the different membranes was normalized to the area of a flat box containing the dataset. A value of 1 signifies a completely flat membrane, and 2 signifies a membrane surface with twice the surface area of a flat sheet. 2-way ANOVA with post-hoc Tukey's test shows that planarity is significantly different for the different membranes, dSDV-PM $p = 0.0000703$***, pSDV-dSDV $p = 0.0006848$***, pSDV-PM $p = 0.0217741$*. See also Supplementary Table S3. **F** Details from the center regions of valve SDVs (The datasets for stages II–IV are the same as in Fig. 2, a different dataset was used for Stage I). The yellow arrowheads highlight putative molecular tethers that connect the plasma membrane to the distal SDV membrane (see Supplementary Fig. S9 for more examples and 3D segmentation). Scale bars are 50 nm. Source data for (**D**, **E**) are provided as a Source Data file.

simulations show that it is likely to miss such events if they are extremely fast (< 1 second), but they have 89% and 99.5% probability of being observed if they last 5 seconds or 10 seconds, respectively (Supplementary Fig. S8). In this regard, a comparison to the fusion of trans-Golgi-derived vesicles with the plasma membrane in yeast that takes about five seconds[37] brings to the conclusion that SDV growth is largely or even entirely independent of material delivery by transport vesicles.

In search for other possible sources for SDV growth, we noticed the close and constant proximity between the valve SDV and the plasma membrane, which was observed in all datasets (Fig. 4A). Quantifying the distances in the segmented volumes revealed a distance of ~20 nm between the distal SDV membrane and the plasma membrane, whereas the proximal SDV membrane moved increasingly farther away from the plasma membrane as the silica thickened (Fig. 4B, D). The constant juxtaposition of membranes over large areas is recognized as a clear indication of organelle-organelle interactions via membrane contact sites that physically tether the two membranes to each other by protein complexes[38–41]. Indeed, our cryoET data revealed many such putative contact site tethers between the distal SDV membrane and the plasma membrane (Fig. 4F and Supplementary Fig. S9). This unexpected observation not only explains SDV positioning precisely at the site where valve exocytosis needs to occur to guarantee proper cell development, but it also provides a possible mechanism for valve SDV growth via the delivery of lipids and proteins from the plasma membrane through contact sites.

At the final stage of SDV development, the SDV membrane dilates in preparation for exocytosis. However, up to that point, there is a remarkably close correlation between the contour of the developing silica surface and the course of the SDV membrane (Fig. 4A). To quantify this, we calculated and mapped the membrane local curvature at each vertex of the segmented membrane surfaces (Fig. 4C). The resulting maps demonstrated that during the initial stage, when distinct silica ribs form, both sides of the SDV are curved. However, when the cribrum pore layer is formed, the proximal SDV membrane becomes flatter, while the distal SDV membrane becomes increasingly curved. The same trend is also apparent in a different analysis of membrane curvature. Normalizing the total surface area of the different membranes to the area of a flat box containing the dataset (namely, the deviation of the actual membrane area from a flat sheet) demonstrated that the plasma membrane and the proximal SDV membrane become flatter as silicification proceeds, whereas the distal SDV membrane becomes more curved (Fig. 4E). This structural correlation may indicate a functional role, placing the SDV membrane as a possible driver for silica morphogenesis. A notable exception from the perfect match between SDV membrane curvature and silica contour is the formation of the intricate tube system at the rim of the valve, the fultoportulae, where the SDV membrane is not closely lining all features of the structure (Supplementary Fig. S10).

A characteristic nanoscale feature of *T. pseudonana* silica is the patterns of cribrum pores (pore diameters ~30 nm; Fig. 1B) that were recently shown to depend on specific proteins, dAnk1-3, which are

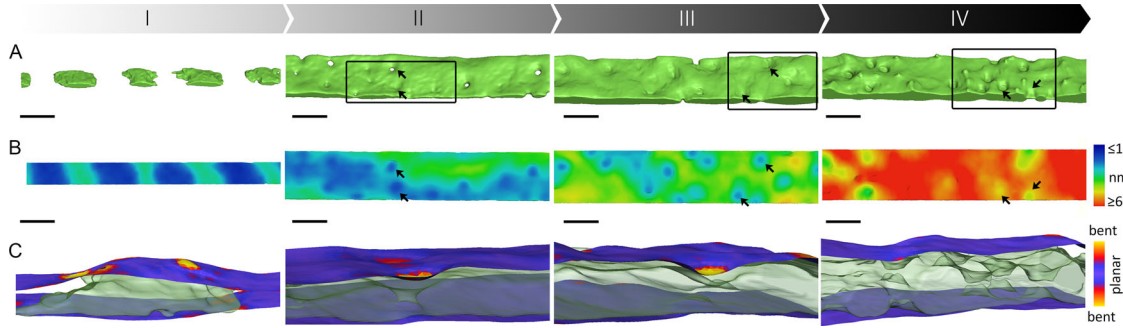

**Fig. 5 | Formation of cribrum pores. A** Surface representation of the silica at different stages of valve SDV development. Some cribrum pores are indicated with arrows. **B** Top view of the distal side of the SDV membrane surface, color-coded according to its distance to the closest location on the proximal side of the SDV membrane. The arrows point to the same positions as in (**A**). Scale bars are 100 nm. **C** Side view of the SDV membranes and silica from the same areas indicated within the black boxes in (**A**). SDV surfaces are color-coded according to the value of the local Gaussian curvature. See Supplementary Movies S1–S4 for animations presenting the entire 3D volumes.

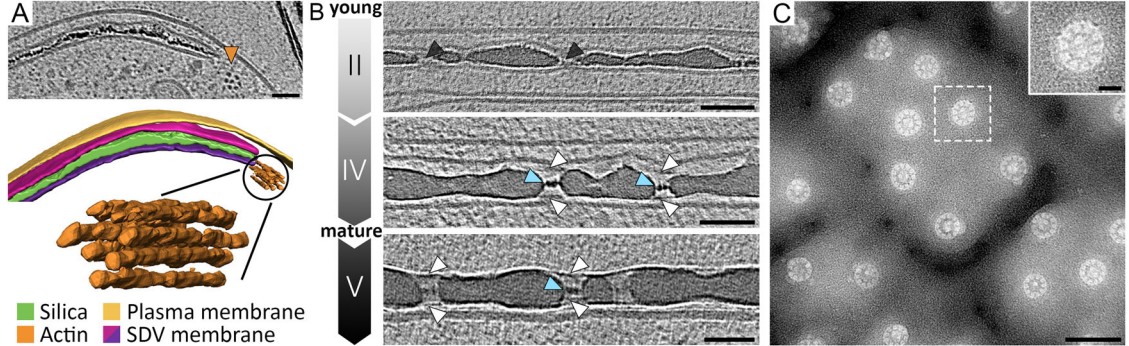

**Fig. 6 | Organic structures associated with silica formation. A** A recurring feature in at least ten cryoET datasets is a bundle of filaments external to the expanding SDV. An orange arrowhead in the image and orange fibers in the surface representation. See also Supplementary Fig. S12. **B** High magnification cryoET slices of the forming silica. At the initial stages, the pores appear empty (black arrowheads), but at a later stage of silica formation, an electron-dense structure blocks the center of the pores (light blue arrowheads), and a thin layer is covering the entire silica (white arrowheads). **C** A representative cryoTEM image of an extracted valve SDV representing two independent experiments. The inset shows one of the cribrum pores in higher magnification. Scale bars are 50 nm and 10 nm for the inset in (**C**).

associated with the cytoplasmic surface of the valve SDV membrane[6]. Formation of the cribrum pores becomes evident from Stage II on, when the inter-rib spaces are filled with silica (Fig. 5A). Mapping of the membrane curvature revealed that the distal and proximal sides of the SDV membrane were bending toward each other in every position where a cribrum pore formed (Fig. 5C). This inward indentation was maintained until the last stage of valve morphogenesis (Fig. 5C). Analyzing the distance between the two sides of the SDV membrane at different stages of valve formation revealed that the pores were always present in local minima of the distances between the two sides of the SDV membrane (Fig. 5B). As the silica thickened, the membrane distance at the pore locations increased (Fig. 5B) to allow for the thickening of the pore walls. These data clearly demonstrated a close correlation between the emergence of a pattern of nanoscale indentations in the SDV membrane and the formation of the pattern of cribrum pores.

The cytoskeleton was repeatedly postulated as a possible player in silica morphogenesis via interactions with the SDV[42]. Even though microtubules were readily detected in our datasets, they were never associated with the SDV (Supplementary Fig. S11). However, a bundle of filaments arranged as a ring at the periphery of the SDV was consistently present (Fig. 6A and Supplementary Fig. S12). The diameter of each filament was 6.8 ± 1.0 nm, which is similar to the diameter of actin fibers[43]. This observation is in accordance with previous suggestions that actin is involved in controlling radial expansion of the SDV[42,44]. Considering the suggested presence of organic templates inside the SDV[1,2,8], in datasets corresponding to Stages I–III, we did not see any

evidence for the presence of biomolecular complexes in the silica-free regions of the SDV lumen. However, at the final stages of silicification (corresponding to Stage IV), the entire valve surface became covered with a ~1 nm thick layer on both sides (Fig. 6B). This layer remained associated with the valve after exocytosis (Stage V) and might represent the insoluble organic matrix that has previously been isolated from the cell walls of *T. pseudonana* and other diatom species[16,45–47]. At Stage IV, the cribrum pores that had been open in the previous stages became clogged by an electron-dense structure (Fig. 6B). When imaging extracted SDVs with cryoTEM, perpendicular to the pore axis, it became apparent that each pore contained a wheel-like structure. The structure consisted of a central ring with eleven spokes connecting the ring to the pore wall. The central ring was generally partitioned into two equal halves by linear structure (Fig. 6C and Supplementary Fig. S13). Hierarchically patterned pores are a hallmark of diatom silica, yet the extremely small feature sizes observed here are unique, to the best of our knowledge (pores with 5 nm diameter, 1 nm wide spokes). It is yet unclear whether the structures inside the cribrum pores are organic or silica-based and what might be their biological function.

## Discussion

This work uses the advanced abilities of cryoET to investigate the structural aspects of silica formation inside the SDV in a direct and quantitative manner. We did not observe any structural features that fit the previously postulated organic templates for silica formation[14,42,48], neither inside nor outside the SDV. Therefore, our structural results do not support the previous scenarios for template-mediated or

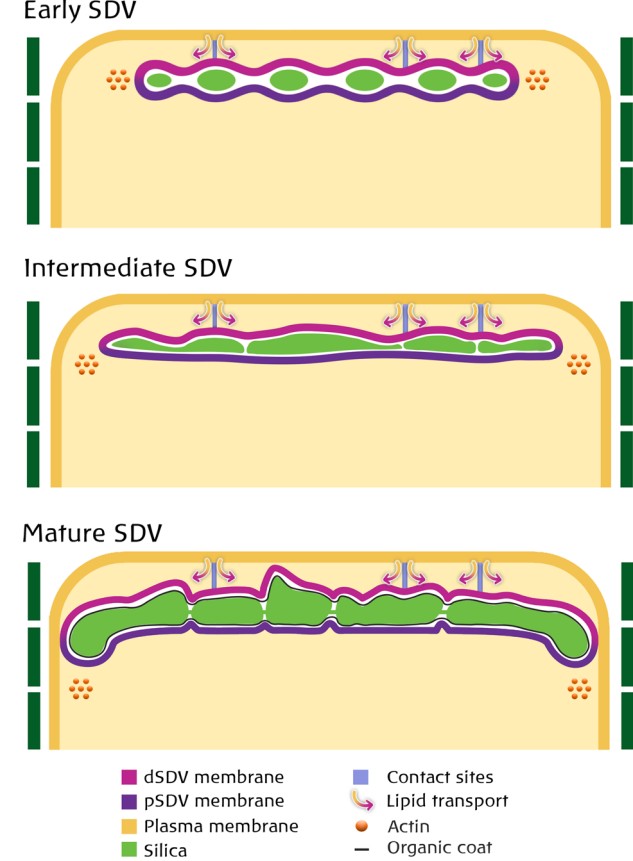

**Fig. 7 | Structural traits of SDV formation and morphogenesis.** The schematic depicts our model of how the tethering of the plasma and SDV membranes can supply the needed lipids and proteins for SDV expansion while creating a different biophysical environment for the distal and proximal sides of the SDV, mirrored in the silica. The peripheral actin ring generates the force for SDV expansion. The initial silica ribs (Early SDV) develop into a continuous mesh of patterned silica (Intermediate SDV), which upon completion contains all silica ornaments, as well as thin layers surrounding the entire valve and in the middle of the cribrum pores (Mature SDV).

template-independent silica morphogenesis. The most prominent observation here is the tight involvement of the SDV membrane in silica morphogenesis and its tethering to the plasma membrane via membrane contact sites (Fig. 4). The widest definition of membrane contact sites is stable and functional locations where two organelles are juxtaposed closer than 60 nm from each other[40]. Our results clearly show such an association between the distal SDV and the plasma membrane (Fig. 4A, B), and putative membrane contact site tethers are also easily identified in the datasets (Fig. 4F and Supplementary Fig. S9). In other organisms, it is known that lipid transfer domains are widely present in contact site proteins, whose function is to shuttle lipids between organelles[49]. In addition, the physical tethering immediately explains how the SDV is positioned exactly at the site where valve exocytosis needs to occur. The cryoET data by itself cannot directly prove a functional role for the contact sites. Nevertheless, since membrane contact sites are known as major factors in inter-organelle communication, future investigations may elucidate their involvement in SDV functions.

Membrane curvature is shown here to be associated with the formation of the cribrum pores. This might be related to the recently identified cytoplasmic proteins, dAnk1-3, that control pore patterns[6]. The proposed model for the dAnk function assumed their interactions with yet unidentified SDV transmembrane proteins that are believed to control the formation and stability of silica-free nanodroplets inside

the SDV[6]. Our data can neither confirm nor exclude the presence of such droplets. However, the data indicate that the machinery that communicates between dAnks on the cytoplasmic side of the SDV membrane and pore patterning in the SDV lumen also includes control over the local membrane curvature.

The finding that the SDV membrane structure is almost synonymous with silica morphogenesis allows us to suggest a model that links silica formation with membrane biophysics (Fig. 7). In this proposition, the SDV expands by the addition of lipids directly from the plasma membrane via the activity of membrane contact site proteins. This means that the distal side of the SDV membrane experiences very different biomechanical conditions from its proximal side because of the tethering between the distal SDV and plasma membrane. Physical tethering was shown to control the membrane curvature of organelles[50], possibly by the generation of strains and forces that deform the membrane[51]. Such a mechanism explains the difference in valve morphology between the proximal and distal sides by virtue of the physical properties of the sur-rounding SDV membrane.

Another possible function of membrane contact sites is the transport of membrane proteins between organelles[52]. Such a role aligns with a previous study on an SDV-membrane protein, Silicanin-1[53], which was localized to intracellular particles prior to its incorporation into the SDV membrane. Even though it is possible that vesicles of the secretory pathways that are loaded with Silicanin-1 directly fuse with the SDV, a more probable scenario based on our data is that they first fuse with the plasma membrane and then are selectively shuttled into the SDV via contact sites.

The actin ring at the periphery of the SDV might be the source of a mechanical force that is needed for SDV expansion. Such activity of expanding actin fibers would trigger a precise flux of protein and lipid transport via the contact sites to maintain the tight lining of the SDV membrane around the forming silica. Alternatively, two simultaneous regulatory pathways might coordinate SDV expansion and membrane fluxes in parallel. In both cases, the proposed role of the membrane contact sites in shuttling lipids and proteins for SDV membrane expansion is in agreement with the suggested functions of contact sites in other systems[40,49].

An interesting concept proposed as a possible mechanism for silica morphogenesis is based on the phase separation of silica-rich and silica-poor phases within the SDV[15]. The chemical and physical sce-narios in this model were speculative, but our quantitative data on the confined volume of the SDV provide some constraints on such pro-cesses. In a phase separation reaction, an initial homogenous phase gives rise to two distinct phases with clear volume fractions that are controlled by the phase diagram. If such a process was continuously happening in the expanding SDV, it would result in a constant volume ratio between the silica and non-silicified volumes of the SDV. Since our data show that only the SDV volume is growing and the ratio between silica and non-silica increases (Fig. 3), we can dismiss such a simplistic scenario. However, other phase separation processes can still be relevant. Possible scenarios are: (i) a liquid-liquid phase separation of macromolecules that form nanodroplets during pore formation, or (ii) the spontaneous formation of patterns of distinct domains within the SDV membrane that direct mineralization.

Lastly, we want to highlight what might be the most enigmatic aspect in the face of the data presented here: the route for Si transport. The tight tethering of the plasma and distal SDV membranes makes the proximal side of the SDV the plausible site for import of soluble silica building blocks from the intracellular Si stores[32,33], across the proximal SDV membrane (Fig. 7). This is counterintuitive to the current and previous observations that as the valve thickens, the added silica is appended to the distal side of the valve (Fig. 3B and Supplementary Fig. S5). The only explanation we can propose is that the cribrum pores serve as conduits for silica building blocks that, for some reason,

diffuse within the SDV from the proximal to the distal side before precipitating as solid silica.

In summary, native-state imaging of the diatom SDV gives structural data that allow to constrain some hypotheses about silica morphogenesis and generate new ones. Remarkably, the data do not provide any evidence to support the involvement of any type of template in the formation of diatom silica, and it limits any template-independent process to a highly confined volume that is growing by adding inorganic material. On the other hand, the SDV membrane emerges from this study as an active player that molds the shape of the forming silica rather than being just a bystander sac delineating the silica deposition volume. We propose that this role is executed by membrane contact sites that tether the SDV to the plasma membrane. Membrane tethering can provide the transport of lipids and SDV-membrane proteins needed for SDV growth. Expansion of the peripheral actin ring may provide the physical driving force for SDV expansion. This scenario, in concert with the membrane transport activity of the contact sites, may establish distinct biophysical conditions between the different sides of the SDV that become imprinted into the silica.

## Methods

### Cell cultures

*Thalassiosira pseudonana* (CCMP1335) cultures were grown in filtered natural Mediterranean seawater with salinity corrected to 3.5%, supplemented with f/2 nutrient recipe and 330 μM silicic acid. Cultures were maintained at 18 °C under 16/8 h light/dark cycles.

### Cell cycle synchronization

*T. pseudonana* synchronization was done using Si starvation, as previously described[13]. The culture was maintained in an exponential growth phase under 12/12 h light/dark cycles by diluting (1:10) into fresh medium every other day during the week before culture synchronization. To induce Si starvation, 100 ml aliquots of culture were centrifuged at $3000 \times g$ for 10 min and resuspended in Si-free artificial seawater or filtered seawater (with < 3 μM Si); this step was repeated three times. The cultures (~ 0.5 million cells/ml) were then grown in the dark for 12 h under agitation in a Si-free medium to arrest the cell cycle. Then, cultures were transferred to continuous light for 4 h of Si starvation. At the end of the Si starvation period, cells were concentrated to about 7–13 million cells/ml, and Si was replenished to 330 μM. PDMPO [2-(4-pyridyl)−5-((4-(2-dimethylaminoethyl-amino-carbamoyl)methoxy)-phenyl)oxazole] (Thermo Fisher Scientific, Waltham, MA, USA) was added to track the formation of new silica. PDMPO fluorescence was monitored by imaging the cultures with an epifluorescence microscope (Nikon Eclipse Ni-U, ex: 365 em: 525). The highest amount of dividing *T. pseudonana* cells containing SDV was counted after 3 hours.

### Plunge freezing

Synchronized *T. pseudonana* cells were vitrified by plunge-freezing on glow discharged 200 mesh copper R2/1 holey carbon film grids (Quantifoil Micro Tools GmbH, Grossloebichau, Germany). In a Leica EM GP (Leica Microsystems GmbH, Wetzlar, Germany), 1 μl of artificial seawater was pipetted on the copper side to enhance media flow to the blotting paper, and 4 μl of cell suspension at $7–13 \times 10^6$ cells/ml was pipetted on the carbon side. The grids were blotted for 8 s from the back side of the grid before they were plunged into a liquid ethane bath cooled by liquid nitrogen.

### Cryo-FIB milling

Plunge-frozen grids containing cells were clipped into auto-grid rims (ThermoFisher Scientific) and loaded into the Zeiss Crossbeam 550 FIB/SEM dual beam microscope (Zeiss, Germany) using a Leica transfer station (VCM EM) equipped with a cryo stage. To create a low-humidity chamber in the transfer station, a plastic cover was used to close it off, and prior to grid handling, $N_2$ flow was used to purge the chamber.

Prior to milling, the grids were coated with organometallic platinum (trimethyl(methylcyclopentadienyl)platinum(IV)) by an in situ Gas Injection System (incorporated in the Zeiss chamber). Depositing a protective organometallic layer prevents cell surface erosion at the milling edge, the major origin of curtain formation. Despite this precaution, curtaining could not be completely avoided due to differences in milling rates between silica-rich versus organic regions. Lamellae were milled in cells that were oriented on their girdle bands. Usually, the lamella width was confined to a few micrometers to span a single cell while keeping its mature valves at the sides intact so the lamellae are well supported. Stress-relief cuts were milled on each side of the lamellae. The lamellae were milled at a 12° tilt relative to the grid plane with the rough milling (to 1 μm thickness) involving two steps using the gallium ion beam at 30 kV and a current of 300 pA and 100 pA. In the final polishing steps, rough-milled lamellae were thinned to 200 nm at a current of 50 pA. Several measures were taken to keep the lamellae free from redeposited material. First, the lamellae on each grid were prepared on a single day, and the grid was removed from the Crossbeam directly after polishing. Second, polishing for all lamellae was done at the very end of the day, after completion of rough milling, in the order starting with the lamella closest to the FIB. Finally, we use the "clean cross-section" approach at the polishing steps, removing slice by slice from the lamella body rather than "back-and-forth" rectangle ablation. These measures reduce the deposition of milling material and ice crystals on the final lamellae surface.

Some lamellae were prepared at the cryo-EM facilities at the UK National Electron Bio-Imaging Center (eBIC). Plunge-frozen grids containing the cells were loaded into a Scios DualBeam system (Thermo-Fisher Scientific) using a Quorum cryo transfer station (PP3010T) equipped with a cryo stage cooled to −168 °C. Prior to milling, grids were sputter coated with platinum in the PP3010T chamber and then coated with a layer of organoplatinum (trimethyl(methylcyclopentadienyl)platinum(IV)) using the gas injection system (incorporated in the Scios chamber). Overall, the milling strategy was the same as described above, with lamellae preparation using a 30 kV $Ga^+$ beam and currents starting at 300 pA (nominal) to 62–72 pA (actual) at the end. Lamellae were prepared using rectangle milling patterns to allow live monitoring of milling progress with the scanning electron microscope operated at 2 kV and 13 pA. The final lamella thickness was less than 300 nm.

### Cryo-electron tomography, volume reconstruction, and volume rendering

Cryo-electron tomography data were collected from 47 pairs of cells. The data sets in Fig. 2 and Supplementary Figs. S2 and S3 were ordered according to SDV maturation, where early stages were interpreted based on SDV lateral growth progress and later stages from silica thickness and fultoportulae completion. The tilt series were acquired using a Titan Krios G3i TEM (Thermo Fisher Scientific, Waltham, MA, USA), operating at 300 kV. Tilt series were recorded on a K3 direct detector (Gatan Inc., Pleasanton, CA, USA) installed behind a Bio-Quantum energy filter (Gatan Inc., Pleasanton, CA, USA), using a slit of 20 eV. All tilt series were recorded in counting mode at a nominal magnification of 33,000 × or 53,000 ×, corresponding to a physical pixel size of 0.26 nm or 0.16 nm, respectively, using the dose-symmetric scheme starting from the lamella pre-tilt of −12° or 12° (dependent on grid orientation) and with 2° increments to acquire ~ 60 tilts for each tomogram[54]. Tilt series were taken at a defocus range of 2–7 μm, using a Volta Phase Plate or an objective aperture of 100 μm inserted. Tilt series were acquired using an automated low-dose procedure implemented in SerialEM v3.8 with a total dose of 100 to 120 e⁻/$Å^2$[55]. Tilt series acquired in the cryo-EM facilities at the UK national electron Bio-Imaging Center (eBIC) were taken with K3, operated in CDS mode with a slit width of 20 eV at a nominal magnification of 42,000 × corresponding to a physical pixel size of 0.22 nm, 13 frames were recorded for each tilt angle.

Tomographic reconstructions were done using either IMOD[56] or AreTomo software[57]. MotionCor2 software was used for frame alignment and averaging in datasets collected as movie frames[58]. The tomograms presented in Fig. 4F and Supplementary Fig. S9 were denoised using CryoCARE[59] or SIRT-like filter in IMOD. In preparation for Cryo-CARE denoising, motion-corrected frames were split into odd and even stacks and averaged separately. Odd and even averages were reconstructed separately using AreTomo, followed by Cryo-CARE denoising. The displayed tomographic slices represent thicknesses of 4–12 nm. Amira software v2021.2 was used for segmentation (Thermo Fisher Scientific, Waltham, MA, USA). Membranes were segmented using the membrane enhancement filter module and manual refinement[60]. Volume measurements were performed using the Label Analysis module of AMIRA 3D software v2021.2. Contact site segmentations were done manually.

### Curvature calculation

The Gaussian curvature calculation was calculated using the Curvature module of Amira 3D software v2021.2. This module computes curvature information for a discrete triangular surface by locally fitting the surface quadratic approximation around each triangle. The output for each triangle is a scalar that specifies the surface geometry. Positive values (parabolic geometry) and negative values (hyperbolic geometry) indicate a surface that bends and, therefore, was colored similarly in the presented surface color code. As curvature estimation is unreliable close to the surface border (edge artifact), these values were removed from the presented analyses. The computed curvature is sensitive to the density and smoothness of the surface; therefore, surfaces were simplified to contain an order of magnitude fewer triangles relative to the original resolution of the data.

### Distance measurements

The distance analysis was performed using the Surface Distance module of Amira 3D software v2021.2. This module measures the distance between two triangulated surfaces. For each vertex on one surface (for example, the SDV membrane surface), the module computes the closest point on the other surface (for example, the plasma membrane surface). The module's output is the magnitude of vectors from each point on one surface to their associated closest points on the other surface.

### Statistical simulations and analyses

The effect of maturation stages on silica/silica-free volumes was tested with a linear regression, separately for silica and silica-free. Differences between dSDV distance datasets and between pSDV distance datasets are described using Cliff's Delta (non-parametric effect size). A zero value indicates complete overlap, and a value of 1 indicates no overlap. Differences in planarity between membranes were tested with 2-way ANOVA, with membrane and stage as categorical factors, followed by Tukey's post-hoc test. All statistics were done using R, v. 4.3.1. Annotated R code for the simulations is available in the supplementary information.

### The possibility that SDV expansion happens via vesicular transport

The putative scenario that simulates the possibility that SDV expansion happens via vesicular transport is based on the coalescence of many small membrane vesicles[30,48], and uses the following assumptions:

### Estimation of the number of vesicles required to form the SDV

1. **Surface area of the mature valve SDV membrane ≈ surface area of a cylinder with a diameter of 4 μm (r = 2 μm) and height of 200 nm (h = 0.2 μm).**

$$SA_{cylinder} = 2\pi rh + 2\pi r^2 \approx 27.65\,\mu m^2$$

2. **Estimation of membrane surface area of coalescing vesicles.** Based on reports of SDV-associated vesicles with diameters ranging from 16 to 80 nm, the surface area of such vesicles was calculated by approximating their geometry to that of a sphere (r = 0.015 μm):

$$SA_{sphere} = 4\pi r^2 \approx 0.0028\,\mu m^2$$

3. **Estimating the number of vesicles fusing to form the SDV.** Based on the previous calculations, it follows that the number of vesicles required to form the entire SDV membrane is:

$$SA_{SDV}/SA_{vesicle} = \text{Number of vesicles}$$
$$= 27.65/0.0028 \approx 9778\text{ vesicles}$$

### Chance of observing putative vesicles in cryo-electron tomogram. Temporal aspect

4. **Duration of a fusion event for a vesicle of 30 nm.** 1, 5 or 10 s

5. **Total duration of SDV expansion.** Based on live-cell imaging data, we know that the formation of a new valve in *T. pseudonana* takes about 90 minutes, while SDV extension happens along the first 30 min[53,61].

### Spatial aspect

6. **Calculating the chance of catching a vesicle in a tomogram.** In addition to the temporal factors in sections 4 and 5 that influence the chances of observing a fusing vesicle, there is a spatial factor: the chance of detecting such a vesicle in a random tomogram volume. We can approximate the probability p of detecting a vesicle with area x in a random tomogram with area y taken of an SDV with area z as:

$$p = \frac{x}{z} \cdot \frac{y}{z}$$

$SDV_{area} = \pi\,r^2 \approx 12.57\,\mu m^2$
$Tomogram_{area} \approx length * depth \approx 1.5 * 0.2 \approx 0.3\,\mu m^2$
$Vesicle_{area} = \pi\,r^2 \approx 0.00071\,\mu m^2$

### Sample preparation for SEM using critical point drying (CPD)

Exponentially growing cells were fixed in a solution of 2% glutaraldehyde and 4% paraformaldehyde in artificial seawater for 1 h at room temperature while shaking. After three washes with deionized water (Milli-Q® IQ 7003 Ultrapure Lab Water System, Merck, Darmstadt, Germany), the cells were dehydrated by washing in a graded ethanol series. The final wash was done in 100 % anhydrous ethanol overnight. The dehydrated samples were dried using liquid $CO_2$ as transitional fluid in a critical point dryer. Dried cells were placed onto a conductive carbon tape on an aluminum stub.

### Sample preparation for SEM using bleach

Exponentially growing cell cultures were centrifuged at $3000 \times g$ for 5 min and washed twice with MilliQ-grade water. Then, cells were re-suspended in 0.06 % sodium hypochlorite and incubated at 4 °C for 2 hr, with agitation. Cells were washed in MilliQ-grade water, pipetted onto track-etched membranes (Sigma-Aldrich, Saint Louis, MO, USA), and placed onto conductive carbon tape on an aluminum stub.

### SEM imaging

Samples were sputter-coated with 2.5 nm iridium (Safematic GmbH, Zizers, Switzerland) and imaged with an Ultra 55 FEG scanning electron microscope (Zeiss, Oberkochen, Germany), using 3 kV, an aperture size 30 μm and a working distance of about 3–5 mm.

## CryoTEM analysis of extracted valve SDVs

Growth of *T. pseudonana* cells and the preparation of cell lysates were performed as described previously[13,26,62]. The lysate was spotted on a copper grid and analyzed as described below.

Both sides of a copper grid with a perforated carbon support film were glow-discharged in a plasma cleaner (FEMTO, Diener electronic GmbH & Co., Ebhausen, Germany) for 12 s. The grid was preincubated with 4 µL of a gold nanoparticles suspension (Ø 10 nm, in PBS) for 60 s. The particles were prepared as follows. The gold particles were resuspended in 100 µL PBS, vortexed for 3 min, sonicated for 5 min, again vortexed for 3 min, and finally centrifuged for 30 s at 500 × *g*. The particles were removed from the grid with a piece of filter paper. Subsequently, a 9 µL aliquot of the cell lysate was spotted onto the grid and incubated for 20 min in a humidity chamber. For plunge freezing, 5 µL of the sample was removed, and the grid was inserted into an automatic plunge freezer that was set to a humidity of 99 % and 18 °C. Within the plunge-freezer, 1 µL of the gold particle suspension was added to the remaining 4 µL of the cell lysate and incubated for 20 s. Subsequently, the sample side of the grid was blotted against a filter paper for 5 s and immediately plunged into liquid ethane. Cryo transmission electron tomography was done on a Titan Halo FEG-TEM instrument (Thermo Fisher Scientific, Waltham, MA, USA) with a single tilt liquid nitrogen cryo holder. The TEM instrument was equipped with a Gatan Image Filter Quantum LS/976 with a K2 Summit direct detection camera (Gatan Inc., Pleasanton, CA, USA). All images were zero-loss filtered with 20 eV slit-width and captured with the program SerialEM in the low-dose mode while the sample was kept at the lowest temperature possible and without counter-heating in the holder. Used nominal magnifications were between 8700 and 130,000. The exposure time was between 1 and 20 s. In each case, 10–20 subframes were taken and automatically aligned by SerialEM to remove potential drift during acquisition. The electron dose was kept between 6 and 8 electrons per pixel and second, and the overall dose for each image was below 100 electrons per square angstrom. The defocus was between − 3.5 µm and − 0.75 µm depending on the magnification.

## CryoEDS measurements

Plunged frozen cells were inserted into a Zeiss Crossbeam 550 FIB/SEM dual-beam microscope (Zeiss, Oberkochen, Germany). Half of the cell was milled at a 17° tilt relative to the grid plane using rough milling with the gallium ion beam at 30 kV and a current of 100 pA. Cryo EDS measurements were performed on the exposed surface in the middle of the cell using a Bruker Quantax microanalysis system with an XFlash 60 mm detector (Bruker, Billerica, MA, USA). EDS maps were acquired with an electron beam acceleration voltage of 8 kV, a working distance of 5 mm, and a probe current of 500 pA. EDS spectra were recorded until significant signals were recorded (~ 4 min).

## Reporting summary

Further information on research design is available in the Nature Portfolio Reporting Summary linked to this article.

## Data availability

Electron microscopy raw data are available through the Dryad digital repository at: https://doi.org/10.5061/dryad.kkwh70s9t. Source data are provided in this paper.

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

## Acknowledgements

We are grateful to Yael Shilderman for the image analysis work, as well as Tobias Fürstenhaupt (MPI-CBG Dresden, Germany) for help with cryoTEM analysis of extracted valve SDV. This project has received funding from the European Research Council (ERC) under the European Union's Horizon 2020 research and innovation program (grant agreement No. 848339). We acknowledge Diamond for access and support of the cryo-EM facilities at the UK National Electron Bio-Imaging Center (eBIC), proposal BI29609, funded by the Wellcome Trust, MRC, and BBSRC. The work was supported by the Deutsche Forschungsgemeinschaft (DFG) through grants KR1853/8-2 (to N.K.) in the framework of Research Unit 2038 (NANOMEE). This research received support from the Irving and Cherna Moskowitz Center for Nano and Bio-Imaging. D.dH. was supported by the

Sustainability and Energy Research Initiative (SAERI) of Weizmann Institute of Science.

## Author contributions

L.A. and D.d.H. performed experiments, analyzed data, and edited the manuscript. N.V. contributed to the segmentations, 3D visualization, and data analysis. D.d.H. and J.G. performed lamella preparation. L.A., D.d.H., and J.G. performed cryoET data collections. C.H. performed sample preparation and cryoTEM imaging of extracted valve SDVs. R.R. conducted the statistical analyses. K.R. assisted with FIB lamellae preparation and collected EDS data. N.E. assisted with cryoET data collection and analysis. N.K. and A.G. provided supervision, funding acquisition, and revised drafts. A.G. wrote the manuscript with inputs from all authours.

## Competing interests

The authors declare no competing interests.
