## [Peer Review File · Nature Communications]

REVIEWER COMMENTS

Reviewer #1 (Remarks to the Author):

The manuscript by Aram et al utilises state of the art cryo electron tomography and focused ion beam EM to provide unprecedented high resolution structural detail of silica deposition in the diatom *T. pseudonana* without the need for staining or fixation. The study follows the relations between the plasma membrane, distal and proximal SDV membranes and forming silica as the valve matures inside the SDV. The authors conclude that their observations do not lend support to template-organised assembly or a template-independent self-assembly. Rather, they present good evidence for key roles for intimate associations between the plasma membrane and the distal SDV membrane in shaping the growth and patterning of the developing structure.

The images are generally clear and their analysis is thorough, though the data presented are largely correlative and descriptive. Notwithstanding this limitation, I consider that the data present new insights in to the morphological regulation of the fundamentally important biological process of diatom silicification, though the work does raise a number of questions that remain unanswered in this study. The authors should address the following concerns and recommendations:

- (line 168) It is stated that no vesicle fusion events were observed though it is not clear how these were investigated. I agree that the absence of vesicle fusion is consistent with the model proposed whereby silica building blocks may be transported through cribrum pores to the distal side of the SDV. However, given that the rate of vesicle fusion in diatoms is not known it is quite feasible that they occur faster than the 5s quoted for yeast vesicle fusion. I find it hard to imagine that vesicle fusion events do not occur in these cells. Did the authors look for vesicle fusion events that were not associated with the SDV (e.g. Golgi/plasma membrane)?
- The evidence presented for the putative molecular tethers between the plasma membrane and the distal SDV membrane (Fig. 4F) is not terribly convincing, given the general graininess of that inter-membrane space. Given that these structures are central to the hypotheses presented, I do think that clearer structural evidence for their existence would strengthen the manuscript.
- (Lin286-) Related to the above, could the tethering sites simply be a mechanism for positioning the SDV rather than the more sophisticated mechanisms of lipid and/or protein transport proposed?
- The wheel like structures in the cribrum pores are fascinating. Some discussion on their possible composition and function would be welcome.
- The role of the actin ring in SDV expansion has been well documented in a number of studies. However, the current manuscript raises more questions about what determines the pattern of rib growth and finer detail associated with the membrane-membrane contacts. Since knockouts of the

dAnks1-3 proteins have been generated in Kroger's lab and have been shown to result in altered pore patterns, an obvious approach would be to perform cryo-EM analysis on these KO strains to determine how the developmental sequence is altered. This may be beyond the scope of the current manuscript but it would potentially provide important new insights into the molecular mechanisms underlying patterning.

Reviewer #2 (Remarks to the Author):

Aram et al use cutting-edge tomography methods to make a major step forward in our understanding of an important yet cryptic biological process. They provide unprecedented and, frankly, stunning images of diatom cells 'caught in the act' of forming their silicified frustule. This provides a series of insights into the apparent mechanism of silicification, including the absence of observable transport vesicles and biotemplates, the involvement of (probably) actin, and the role of the membrane in dictating silica morphology. The authors do a particularly good job of placing their work in the context of prior studies as well as considering the various hypotheses and ideas that have been proposed to explain this mechanism. There are several new observations here (to my knowledge) including the exciting observation of membrane contact sites, and the work is synthesised into somewhat of a unified model of frustule formation.

This is an observational study. I am not highly expert in tomography techniques but the samples are obviously expertly prepared for imaging and it seems very unlikely the results are affected by preparation artefacts. The data analysis (across multiple samples) is largely qualitative, which is correct in my view. Where more quantitative analyses are performed (e.g. Fig. 4) statistical tests are used appropriately. Methods are provided in full detail sufficient for others to follow.

This work is exceptionally well-presented and well-written and will be clear to a general audience. This will be highly-cited and seen as a cornerstone of the field for many years to come. I strongly endorse accepting this article.

Comments:

The one aspect of the images that is slightly less clear to the untrained eye is the apparent connection between the SDV and plasma membrane (Fig. 4E). This is a novel and potentially

important observation, and is a key aspect of the new model proposed. Is it possible to show these features in an even clearer way?

Line 303/304: The authors say that “the tethering between the distal SDV and plasma membrane can generate strains and forces, giving rise to wrinkling (49).” Please elaborate on the nature and magnitude of these forces for the general reader, citation 49 is not very illuminating on this issue.

Line 350-352: “The tight tethering of the plasma and distal SDV membranes makes the proximal side of the SDV the main site for any transport from the intracellular Si stores (Scheme 1)”. Perhaps I have misunderstood, but since the authors do not observe transport vesicles, does this bring into question the existence of such stores?

Reviewer #3 (Remarks to the Author):

The manuscript reports on work on the silica formation in the algae *Thalassiosira pseudonana* employing cryo-TEM and cryo-TEM tomography. The authors find that silica formation is not determined by a templating role of the membrane matrix and hypothesize that it is driven by membrane tethering based on an assumed time-sequence constructed from sets of cryo-TEM data taken from a selection of algae cells.

Albeit being interesting in the context of understanding silica based mineralising systems and some interesting data sets showing details of the cell wall membrane and its association with the mineralising phase the paper is lacking break-through insights of the actual mechanism of silica formation in these algae on a level that would merit publication in *Nat. Comm.* The authors convincingly show that the mineralisation does not occur via a rigorous templating pathway. However, although their argumentation is not necessarily invalid, it lacks sufficient underpinning by the presented data elaborated in the following.

- The cryo-TEM image series provided in the Supplementary data does not allow to determine the actual location of imaging used for the evaluation. Are there any SEM data of those samples prepared by FIB for the cryo-TEM and tomography and identification of positions on the cells used for the evaluation?

- Consequently it appears relatively random to group cryo-TEM images with respect to the membrane thickness and then associate this with different stages of growth. How can it be excluded that the thickness variation are not dominated by random positional variations within the organism? It remains to speculative to derive formation dynamics from static images in this manner.

- What is the original size distribution of the individual SDVs used for the investigation? It is unclear how the features used for evaluation are related to the actual SDV size.

- How was the potential impact of the FIB milling on observable structural features determined/excluded as playing a role in affecting the results?

- The assignment of different layers in the cryo-TEM images/tomograms are solely down to association of image contrast. A compositional analysis of the FIB samples would be recommendable to ensure that at least the Si containing layers is properly identified. It would also provide information on the stoichiometrical variation of Si content in these layers.

Response to reviewer 1

Reviewer #1 (Remarks to the Author):

The manuscript by Aram et al utilises state of the art cryo electron tomography and focused ion beam EM to provide unprecedented high resolution structural detail of silica deposition in the diatom *T. pseudonana* without the need for staining or fixation. The study follows the relations between the plasma membrane, distal and proximal SDV membranes and forming silica as the valve matures inside the SDV. The authors conclude that their observations do not lend support to template-organised assembly or a template-independent self-assembly. Rather, they present good evidence for key roles for intimate associations between the plasma membrane and the distal SDV membrane in shaping the growth and patterning of the developing structure.

The images are generally clear and their analysis is thorough, though the data presented are largely correlative and descriptive. Notwithstanding this limitation, I consider that the data present new insights in to the morphological regulation of the fundamentally important biological process of diatom silicification, though the work does raise a number of questions that remain unanswered in this study. The authors should address the following concerns and recommendations:

We appreciated the constructive and positive feedback of the reviewer.

- (line 168) It is stated that no vesicle fusion events were observed though it is not clear how these were investigated. I agree that the absence of vesicle fusion is consistent with the model proposed whereby silica building blocks may be transported through cribrum pores to the distal side of the SDV. However, given that the rate of vesicle fusion in diatoms is not known it is quite feasible that they occur faster than the 5s quoted for yeast vesicle fusion. I find it hard to imagine that vesicle fusion events do not occur in these cells. Did the authors look for vesicle fusion events that were not associated with the SDV (e.g. Golgi/plasma membrane)?

Indeed, vesicular transport is present and detected in our dataset in various cellular contexts that are not SDV-related. We agree that including such information will help to interpret the observation that no such events were observed in association with the SDV, so we included a new supporting figure (the new Fig S7) that shows several cases of vesicular transport events. The reader is referred to this in lines 170-172: "Here, we did not observe a single event of a vesicle fusing with the valve SDV, even though such events are readily visible with cryoET in other biological systems,34–36 and also in our datasets in other cellular contexts (Fig. S7)."

- The evidence presented for the putative molecular tethers between the plasma membrane and the distal SDV membrane (Fig. 4F) is not terribly convincing, given the general graininess of that inter-membrane space. Given that these structures are central to the hypotheses presented, I do think that clearer structural evidence for their existence would strengthen the manuscript.

We agree that the contrast between the contact sites and their environment is low. We took several measures to improve this point:

1) We analyzed the presented datasets according to the latest standards in the field and revised panel F in Fig. 4. Now the contrast is improved for better visualization of the contact sites.

2) the new Fig. S9 contains 6 other examples for contact sites in the datasets, as well as two 3D segmentations of their occurrence. These examples should help the reader to evaluate the structural evidence. The reader is referred to this supporting figure in the caption of Fig. 4, in line 186 and 294.

3) We added in line 185 a reference (number 41) to a recent paper (Wozny et al. Nature 2023), in which the authors study contact sites with cryoET. We note that even though this work took a much more detailed structural approach, the images of the contact sites are qualitatively comparable to our data (see Figure 2 in their paper).

- (Lin286-) Related to the above, could the tethering sites simply be a mechanism for positioning the SDV rather than the more sophisticated mechanisms of lipid and/or protein transport proposed?

Yes, positioning of the SDV is the direct and simple function. This is the first notion that is stated in the last sentence of this paragraph in lines 186-189: "This unexpected observation not only explains SDV positioning precisely at the site where valve exocytosis needs to occur to guarantee proper cell development, but it also provides a possible mechanism for valve SDV growth via the delivery of lipids and proteins from the plasma membrane through contact sites."

- The wheel like structures in the cribrum pores are fascinating. Some discussion on their possible composition and function would be welcome.

These structures are very interesting but unfortunately, we cannot provide further information about their biochemical nature. Nevertheless, we added a new supporting figure (Fig. S13) that contains new 3D information of their structure as segmented from the cryoET data. This 3D structure suffers from the 'missing wedge' effect that makes it look anisotropic, but based on the reviewer's comment we believe it can be important to some readers. The reader is referred to this data in line 267.

Since we do not have an enough data for an informative discussion we included the following sentences as a minor discussion-like statement in lines 268-271: "Hierarchically patterned pores are a hallmark of diatom silica, yet the extremely small feature sizes observed here are unique, to the best of our knowledge (pores with 5 nm diameter, 1 nm wide spokes). It is yet unclear whether the structures inside the cribrum pores are organic or silica-based and what might be their biological function.

- The role of the actin ring in SDV expansion has been well documented in a number of studies. However, the current manuscript raises more questions about what determines the pattern of rib growth and finer detail associated with the membrane-membrane contacts. Since knockouts of the dAnks1-3 proteins have been generated in Kroger's lab and have been shown to result in altered pore patterns, an obvious approach would be to preform cryo-EM analysis on these KO strains to determine how the developmental sequence is altered. This may be beyond the scope of the current manuscript but it would potentially provide important new insights into the molecular mechanisms underlying patterning.

This line of research is active in both groups, but as the reviewer notes, it is a very challenging task that we hope to report on in the future.

Response to reviewer 2

Reviewer #2 (Remarks to the Author):

Aram et al use cutting-edge tomography methods to make a major step forward in our understanding of an important yet cryptic biological process. They provide unprecedented and, frankly, stunning images of diatom cells 'caught in the act' of forming their silicified frustule. This provides a series of insights into the apparent mechanism of silicification, including the absence of observable transport vesicles and biotemplates, the involvement of (probably) actin, and the role of the membrane in dictating silica morphology. The authors do a particularly good job of placing their work in the context of prior studies as well as considering the various hypotheses and ideas that have been proposed to explain this mechanism. There are several new observations here (to my knowledge) including the exciting observation of membrane contact sites, and the work is synthesised into somewhat of a unified model of frustule formation.

This is an observational study. I am not highly expert in tomography techniques but the samples are obviously expertly prepared for imaging and it seems very unlikely the results are affected by preparation artefacts. The data analysis (across multiple samples) is largely qualitative, which is correct in my view. Where more quantitative analyses are performed (e.g. Fig. 4) statistical tests are used appropriately. Methods are provided in full detail sufficient for others to follow.

This work is exceptionally well-presented and well-written and will be clear to a general audience. This will be highly-cited and seen as a cornerstone of the field for many years to come. I strongly endorse accepting this article.

We are grateful for the encouraging words and the appreciation of the work. We used the following comments to further improve the manuscript.

Comments:

The one aspect of the images that is slightly less clear to the untrained eye is the apparent connection between the SDV and plasma membrane (Fig. 4E). This is a novel and potentially important observation, and is a key aspect of the new model proposed. Is it possible to show these features in an even clearer way?

This comment is shared by reviewer 1, and the same measures that we took to improve it are copied here:

1) We analyzed the presented datasets according to the latest standards in the field and revised panel F in Fig. 4. Now the contrast is improved for better visualization of the contact sites.

2) the new Fig. S9 contains 6 other examples for contact sites in the datasets, as well as two 3D segmentations of their occurrence. These examples should help the reader to evaluate the structural evidence. The reader is referred to this supporting figure in the caption of Fig. 4, in line 186 and 294.

3) We added in line 185 a reference (number 41) to a recent paper (Wozny et al. Nature 2023), in which the authors study contact sites with cryoET. We note that even though this work took a much more detailed structural approach, the images of the contact sites are qualitatively comparable to our data (see Figure 2 in their paper).

Line 303/304: The authors say that “the tethering between the distal SDV and plasma membrane can generate strains and forces, giving rise to wrinkling (49).” Please elaborate on the nature and magnitude of these forces for the general reader, citation 49 is not very illuminating on this issue.

We elaborate on the relevance of contact sites to membrane curvature and added a reference to a work that specifically shows how the ER membrane curvature is associated with contact sites (Collado, J. et al. (2019) Tricalbin-Mediated Contact Sites Control ER Curvature to Maintain Plasma Membrane Integrity. *Developmental Cell*, 51(4), 476-487.e7. <https://doi.org/10.1016/j.devcel.2019.10.018>). The revised text in lines 311-315 now reads: “This means that the distal side of the SDV membrane experiences very different biomechanical conditions from its proximal side because of the tethering between the distal SDV and plasma membrane. Physical tethering was shown to control membrane curvature of organelles,50 possibly by the generation of strains and forces that deform the membrane.51”

Line 350-352: “The tight tethering of the plasma and distal SDV membranes makes the proximal side of the SDV the main site for any transport from the intracellular Si stores (Scheme 1)”. Perhaps I have misunderstood, but since the authors do not observe transport vesicles, does this bring into question the existence of such stores?

This sentence refers to the transport of soluble building blocks via membrane transporters and not via vesicle fusion. Such stores are well known in diatoms, even though their exact intracellular location is yet unclear. We revised the sentence to spell this point out (line 363-365): “The tight tethering of the plasma and distal SDV membranes makes the proximal side of the SDV the plausible site for import of soluble silica building blocks from the intracellular Si stores,32,33 across the proximal SDV membrane (Scheme 1).”

Response to reviewer 3

Reviewer #3 (Remarks to the Author):

The manuscript reports on work on the silica formation in the algae *Thalassiosira pseudonana* employing cryo-TEM and cryo-TEM tomography. The authors find that silica formation is not determined by a templating role of the membrane matrix and hypothesize that it is driven by membrane tethering based on an assumed time-sequence constructed from sets of cryo-TEM data taken from a selection of algae cells.

Albeit being interesting in the context of understanding silica based mineralising systems and some interesting data sets showing details of the cell wall membrane and its association with the mineralising phase the paper is lacking break-through insights of the actual mechanism of silica formation in these algae on a level that would merit publication in *Nat. Comm.* The authors convincingly show that the mineralisation does not occur via a rigorous templating pathway. However, although their argumentation is not necessarily invalid, it lacks sufficient underpinning by the presented data elaborated in the following.

We disagree with this reviewer’s opinion regarding new insights into the mechanism of silica morphogenesis for the following reasons:

- We demonstrate that the SDV membrane itself rather than templates in the SDV lumen or outside the SDV determines the morphology of the silica. This is demonstrated by the remarkable congruence of membrane curvature and nascent silica structure at all developmental stages.
- Our data clearly demonstrate that formation of the patterns of the 30 nm sized cribrum pores is a direct consequence of invagination of the SDV membrane. This is different to previous models that postulated templating of cribrum pores by nanodroplets in the SDV lumen.
- Furthermore, the fact that we discovered a lack of vesicle transport to the SDV and the unanticipated presence of contact sites between the SDV and the plasma membrane, provides an entirely new paradigm for the biogenesis of mineral forming vesicles.

Therefore, we used the reviewers' comments to explain more precisely and rigorously how we interpret the acquired data. We do hope that these modifications will convince the reviewer that this work provides a breakthrough in the understanding of the cellular silicification process.

- The cryo-TEM image series provided in the Supplementary data does not allow to determine the actual location of imaging used for the evaluation. Are there any SEM data of those samples prepared by FIB for the cryo-TEM and tomography and identification of positions on the cells used for the evaluation?

We constructed a new supporting figure (the new Fig. S2) in which all the low magnification images of the lamellae are presented and the locations where cryoET data was collected are indicated. This figure also includes examples for FIB-SEM sample preparation. The reader is referred to this figure in line 116 and in the captions of Figs. 2 and 3.

- Consequently it appears relatively random to group cryo-TEM images with respect to the membrane thickness and then associate this with different stages of growth. How can it be excluded that the thickness variation are not dominated by random positional variations within the organism? It remains to speculative to derive formation dynamics from static images in this manner.

It is true that the cryoET samples are static, but careful analysis allows to derive dynamic information. In our case, the only assumption is that silicification involves a monotonous growth of the valve, first expanding laterally and then thickening (This is the prevailing model for the mode of silica growth in the SDV, and has been experimentally demonstrated for *T. pseudonana* (see reference 28)). We present at the bottom of the new Fig. S2 a graph that shows how the ratio of SDV to cell diameter correlates with thicker silica. This analysis shows that our assignment of the four different stages is not random but follows specific traits that are related to the growth process. The reader is referred to this information in lines 114-116: "We created a derived timeline of valve formation within the SDV by arranging the collected datasets in increasing order of SDV diameter, silica thickness, and development of secondary ornaments (Fig. 2, Figs. S2, S3)."

- What is the original size distribution of the individual SDVs used for the investigation? It is unclear how the features used for evaluation are related to the actual SDV size.

We hope the new information discussed in the previous points resolves this issue (see Fig. S2).

- How was the potential impact of the FIB milling on observable structural features determined/excluded as playing a role in affecting the results?

FIB milling is the standard for cryoET sample preparation, and as such it is used in numerous works. The possibility of sample damage was considered by several of the pioneering works of this field. For example, please check:

1) Berger, C., Premaraj, N., Ravelli, R. B. G., Knoop, K., López-Iglesias, C., & Peters, P. J. (2023). Cryo-electron tomography on focused ion beam lamellae transforms structural cell biology. *Nature Methods* 2023 20:4, 20(4), 499–511. <https://doi.org/10.1038/s41592-023-01783-5>

2) Nogales, E., & Mahamid, J. (2024). Bridging structural and cell biology with cryo-electron microscopy. *Nature* 2024 628:8006, 628(8006), 47–56. <https://doi.org/10.1038/s41586-024-07198-2>

- The assignment of different layers in the cryo-TEM images/tomograms are solely down to association of image contrast. A compositional analysis of the FIB samples would be recommendable to ensure that at least the Si containing layers is properly identified. It would also provide information on the stoichiometrical variation of Si content in these layers.

We include cryoFIB-EDS data on a lamella that shows Si signal from the SDV in the new Fig. S1. The reader is referred to this figure in line 94.

REVIEWER COMMENTS

Reviewer #1 (Remarks to the Author):

The authors have addressed all of my earlier concerns in this revised manuscript. I am satisfied that the new information provided as described by the authors in the response to reviewers has allayed my concerns and helped to clarify the issues raised.

Reviewer #2 (Remarks to the Author):

The authors have addressed the issue raised in my previous review. I support publication of this work.

Reviewer #3 (Remarks to the Author):

Here are my comments on the authors responses to the points raised by myself. The individual points are indicated by quotation marks.

“The cryo-TEM image series...”

The additional material provided is sufficient.

“Consequently, it appears relatively random...”

The response is not satisfactory since it claims a uniform growth behavior as a standard on which the growth dynamics from static images is derived. Although overall the described lateral expanding prior to the thickening has been observed it does not allow to specifically claim a local variation of thickness in one sample to be directly representative of the time dependence of the wall growth. Conditions can vary locally over time, specifically as the organelle grows.

“What is the original size distribution...”

The additional material provided is sufficient.

“How was the potential impact of the FIB ...”

I am not happy with the referencing of other general publications on the subject of cryo-FIBSEM preparation in response to the request to identify how sample damage was minimized. Details on e.g. avoiding curtaining and redeposition of milled material is critical to obtain sufficient sample quality so that the imaging and EDS analyses are sufficiently reliable to exclude sample

modifications due to beam/sample interactions. Each sample usually requires specific measures to optimise these conditions.

“The assignment of different layers...”

The EDS maps and spectra show the presence of Si. A remaining question was if there is evidence of a stoichiometrical variation with respect to [Si]/[C]/[O] in the silicate walls.

Response to reviewer 1

Reviewer #1 (Remarks to the Author):

The authors have addressed all of my earlier concerns in this revised manuscript. I am satisfied that the new information provided as described by the authors in the response to reviewers has allayed my concerns and helped to clarify the issues raised.

Thank you for the effort and time invested in the review process.

Response to reviewer 2

Reviewer #2 (Remarks to the Author):

The authors have addressed the issue raised in my previous review. I support publication of this work.

Thank you for the effort and time invested in the review process.

Response to reviewer 3

Reviewer #3 (Remarks to the Author):

Here are my comments on the authors' responses to the points raised by myself. The individual points are indicated by quotation marks.

“The cryo-TEM image series...”

The additional material provided is sufficient.

Thank you.

“Consequently, it appears relatively random...”

The response is not satisfactory since it claims a uniform growth behavior as a standard on which the growth dynamics from static images is derived. Although overall the described lateral expanding prior to the thickening has been observed it does not allow to specifically claim a local variation of thickness in one sample to be directly representative of the time dependence of the wall growth. Conditions can vary locally over time, specifically as the organelle grows.

The reviewer is correct that conditions vary locally and we cannot unequivocally set a growth sequence. However, for this work we only use the assumption that silica growth is **monotonous** (i.e. proceeding only via material addition with no dissolution), and we do not claim **uniform** (i.e. that all valves go through the same exact stages). The previously added data in Fig. S2 demonstrate that our choice of four representative growth stages is in accordance with the overall trend of silica diameter and thickness.

We added a direct reference to this point in the main text, lines 114-119: “We created a derived timeline of valve formation within the SDV by arranging the collected datasets in increasing order of SDV diameter, silica thickness, and development of secondary ornaments (Fig. 2, Figs. S2, S3). Even though this timeline cannot be unequivocally defined, it is robust enough to serve

as a basis to separate the growth process into four representative stages, which were reconstructed for two characteristic locations inside the cell, the central region of the SDV and its growing edge (Fig. 2).”

“What is the original size distribution...”

The additional material provided is sufficient.

Thank you.

“How was the potential impact of the FIB ...”

I am not happy with the referencing of other general publications on the subject of cryo-FIBSEM preparation in response to the request to identify how sample damage was minimized. Details on e.g. avoiding curtaining and redeposition of milled material is critical to obtain sufficient sample quality so that the imaging and EDS analyses are sufficiently reliable to exclude sample modifications due to beam/sample interactions. Each sample usually requires specific measures to optimise these conditions.

Thank you for the clarification on the rationale behind the comment. Now we understand that the request is to elaborate on the measures taken with our samples to avoid milling-related artifacts. This was done by adjusting the milling sequence in both machines used in this work so the final polishing steps are taken with the lowest ion current and at a sequence that minimizes re-deposition.

The sub-section ‘Cryo-FIB milling’ in the Materials section was expanded to include these experimental additions. The major ones are: “To create a low-humidity chamber in the transfer station, a plastic cover was used to close it off, and prior to grid handling N₂ flow was used to purge the chamber... Depositing a protective organometallic layer prevents cell surface erosion at the milling edge, the major origin of curtain formation. Despite this precaution, curtaining could not be completely avoided due to differences in milling rates between silica rich versus organic regions. Lamellae were milled in cells that were oriented on their girdle bands. Usually, the lamella width was confined to a few micrometers to span a single cell while keeping its mature valves at the sides intact so the lamellae are well supported. Stress-relief cuts were milled on each side of the lamellae... Several measures were taken to keep the lamellae free from redeposited material. First, the lamellae on each grid were prepared on a single day, and the grid was removed from the Crossbeam directly after polishing. Second, polishing for all lamellae was done at the very end of the day, after completion of rough milling, in the order starting with the lamella closest to the FIB. Finally, we use the "clean cross-section" approach at the polishing steps, removing slice by slice from the lamella body rather than "back-and-forth" rectangle ablation. These measures reduce deposition of milling material and ice crystals on the final lamellae surface.”

“The assignment of different layers...”

The EDS maps and spectra show the presence of Si. A remaining question was if there is evidence of a stoichiometrical variation with respect to [Si]/[C]/[O] in the silicate walls.

Indeed, we did not provide answer to this question. Unfortunately the answer is that from the cryoEDS data we cannot learn about stoichiometry related to C and O. The reason is that the interaction volume that generates the signal within the sample is at least hundreds of nanometers in the Z direction (a little bit less in the X and Y directions). Therefore, the EDS signal originates both from the silica and the surrounding cellular materials. This precludes direct analysis of silica compositions.

REVIEWERS' COMMENTS

Reviewer #3 (Remarks to the Author):

Here are my comments on the remaining points to be addressed by the authors:

“Consequently, it appears relatively random...”

We added a direct reference to this point in the main text, lines 114-119: “We created a derived timeline of valve formation within the SDV by arranging the collected datasets in increasing order of SDV diameter, silica thickness, and development of secondary ornaments (Fig. 2, Figs. S2, S3). Even though this timeline cannot be unequivocally defined, it is robust enough to serve as a basis to separate the growth process into four representative stages, which were reconstructed for two characteristic locations inside the cell, the central region of the SDV and its growing edge (Fig. 2).”

I still believe that it is overstated to call this a "timeline". It would be more correct to refer to it "as a sequence of images sorted on the basis of the silica thickness." The growth dynamics cannot be inferred from this and the basic assumption remains a matter of speculation.

“How was the potential impact of the FIB ...”

Thank you for the clarification on the rationale behind the comment. Now we understand that the request is to elaborate on the measures taken with our samples to avoid milling-related artifacts. This was done by adjusting the milling sequence in both machines used in this work so the final polishing steps are taken with the lowest ion current and at a sequence that minimizes re-deposition.

The sub-section ‘Cryo-FIB milling’ in the Materials section was expanded to include these experimental additions. The major ones are: “To create a low-humidity chamber in the transfer station, a plastic cover was used to close it off, and prior to grid handling N₂ flow was used to purge the chamber... Depositing a protective organometallic layer prevents cell surface erosion at the milling edge, the major origin of curtain formation. Despite this precaution, curtaining could not be completely avoided due to differences in milling rates between silica rich versus organic regions. Lamellae were milled in cells that were oriented on their girdle bands. Usually, the lamella width was confined to a few micrometers to span a single cell while keeping its mature valves at the sides intact so the lamellae are well supported. Stress-relief cuts were milled on each side of the lamellae... Several measures were taken to keep the lamellae free from redeposited material. First,

the lamellae on each grid were prepared on a single day, and the grid was removed from the Crossbeam directly after polishing. Second, polishing for all lamellae was done at the very end of the day, after completion of rough milling, in the order starting with the lamella closest to the FIB. Finally, we use the "clean cross-section" approach at the polishing steps, removing slice by slice from the lamella body rather than "back-and-forth" rectangle ablation. These measures reduce deposition of milling material and ice crystals on the final lamellae surface."

Adding details of the FIB presentation is commendable. What do you mean by "closest to the FIB"?

"The assignment of different layers..."

The EDS maps and spectra show the presence of Si. A remaining question was if there is evidence of a stoichiometrical variation with respect to [Si]/[C]/[O] in the silicate walls.

Indeed, we did not provide answer to this question. Unfortunately the answer is that from the cryoEDS data we cannot learn about stoichiometry related to C and O. The reason is that the interaction volume that generates the signal within the sample is at least hundreds of nanometers in the Z direction (a little bit less in the X and Y directions). Therefore, the EDS signal originates both from the silica and the surrounding cellular materials. This precludes direct analysis of silica compositions.

The interaction volume could have been optimized using appropriate aperture, intensity and acceleration voltage values also using Z-variations to probe the Z-dependence of the signal to enable much a smaller probe depth.

Response to reviewer 3

Reviewer #3 (Remarks to the Author):

Here are my comments on the remaining points to be addressed by the authors:

“Consequently, it appears relatively random...”

We added a direct reference to this point in the main text, lines 114-119: “We created a derived timeline of valve formation within the SDV by arranging the collected datasets in increasing order of SDV diameter, silica thickness, and development of secondary ornaments (Fig. 2, Figs. S2, S3). Even though this timeline cannot be unequivocally defined, it is robust enough to serve as a basis to separate the growth process into four representative stages, which were reconstructed for two characteristic locations inside the cell, the central region of the SDV and its growing edge (Fig. 2).”

I still believe that it is overstated to call this a "timeline". It would be more correct to refer to it "as a sequence of images sorted on the basis of the silica thickness." The growth dynamics cannot be inferred from this and the basic assumption remains a matter of speculation.

We appreciate the reviewer concerns about the accuracy of the statements regarding how the static data relate to a dynamic process. The current phrasing is delivering the same ideas as the reviewer suggests: “We created a derived timeline of valve formation within the SDV by arranging the collected datasets in increasing order of SDV diameter, silica thickness, and development of secondary ornaments (Fig. 2, Figs. S2, S3). Even though this timeline cannot be unequivocally defined, it is robust enough to serve as a basis to separate the growth process into four representative stages, which were reconstructed for two characteristic locations inside the cell, the central region of the SDV and its growing edge (Fig. 2).”

“How was the potential impact of the FIB ...”

Thank you for the clarification on the rationale behind the comment. Now we understand that the request is to elaborate on the measures taken with our samples to avoid milling-related artifacts. This was done by adjusting the milling sequence in both machines used in this work so the final polishing steps are taken with the lowest ion current and at a sequence that minimizes re-deposition.

The sub-section ‘Cryo-FIB milling’ in the Materials section was expanded to include these experimental additions. The major ones are: “To create a low-humidity chamber in the transfer station, a plastic cover was used to close it off, and prior to grid handling N2 flow was used to purge the chamber... Depositing a protective organometallic layer prevents cell surface erosion at the milling edge, the major origin of curtain formation. Despite this precaution, curtaining could not be completely avoided due to differences in milling rates between silica rich versus organic regions. Lamellae were milled in cells that were oriented on their girdle bands. Usually, the lamella width was confined to a few micrometers to span a single cell while keeping its mature valves at the sides intact so the lamellae are well supported. Stress-relief cuts were milled on each side of the lamellae... Several measures were taken to keep the lamellae free from redeposited material. First, the lamellae on each grid were prepared on a single day, and the grid was removed from the Crossbeam directly after polishing. Second, polishing for all

lamellae was done at the very end of the day, after completion of rough milling, in the order starting with the lamella closest to the FIB. Finally, we use the "clean cross-section" approach at the polishing steps, removing slice by slice from the lamella body rather than "back-and-forth" rectangle ablation. These measures reduce deposition of milling material and ice crystals on the final lamellae surface."

Adding details of the FIB presentation is commendable. What do you mean by "closest to the FIB"?

We now explain that "closest to the FIB" mean the side of the lamella that is closer to the FIB source.

"The assignment of different layers..."

The EDS maps and spectra show the presence of Si. A remaining question was if there is evidence of a stoichiometrical variation with respect to [Si]/[C]/[O] in the silicate walls.

Indeed, we did not provide answer to this question. Unfortunately the answer is that from the cryoEDS data we cannot learn about stoichiometry related to C and O. The reason is that the interaction volume that generates the signal within the sample is at least hundreds of nanometers in the Z direction (a little bit less in the X and Y directions). Therefore, the EDS signal originates both from the silica and the surrounding cellular materials. This precludes direct analysis of silica compositions.

The interaction volume could have been optimized using appropriate aperture, intensity and acceleration voltage values also using Z-variations to probe the Z-dependence of the signal to enable much a smaller probe depth.

We disagree with the reviewer on this point. The spatial resolution is limited by the electron energy that needs to be high enough to excite Si atoms. The probe itself on the sample can be as small as few nm but the interaction volume within the sample that generates the Xray photons is much larger and cannot be directly varied to answer this point.